# Derivation of Highly Predictive 3D-QSAR Models for hERG Channel Blockers Based on the Quantum Artificial Neural Network Algorithm

**DOI:** 10.3390/ph16111509

**Published:** 2023-10-24

**Authors:** Taeho Kim, Kee-Choo Chung, Hwangseo Park

**Affiliations:** Department of Bioscience and Biotechnology, Sejong University, 209 Neungdong-ro, Kwangjin-gu, Seoul 05006, Republic of Korea; tahok@hanmail.net

**Keywords:** hERG channel blockers, 3D-QSAR, structural alignment, molecular ESP descriptor, artificial neural network

## Abstract

The hERG potassium channel serves as an annexed target for drug discovery because the associated off-target inhibitory activity may cause serious cardiotoxicity. Quantitative structure–activity relationship (QSAR) models were developed to predict inhibitory activities against the hERG potassium channel, utilizing the three-dimensional (3D) distribution of quantum mechanical electrostatic potential (ESP) as the molecular descriptor. To prepare the optimal atomic coordinates of dataset molecules, pairwise 3D structural alignments were carried out in order for the quantum mechanical cross correlation between the template and other molecules to be maximized. This alignment method stands out from the common atom-by-atom matching technique, as it can handle structurally diverse molecules as effectively as chemical derivatives that share an identical scaffold. The alignment problem prevalent in 3D-QSAR methods was ameliorated substantially by dividing the dataset molecules into seven subsets, each of which contained molecules with similar molecular weights. Using an artificial neural network algorithm to find the functional relationship between the quantum mechanical ESP descriptors and the experimental hERG inhibitory activities, highly predictive 3D-QSAR models were derived for all seven molecular subsets to the extent that the squared correlation coefficients exceeded 0.79. Given their simplicity in model development and strong predictability, the 3D-QSAR models developed in this study are expected to function as an effective virtual screening tool for assessing the potential cardiotoxicity of drug candidate molecules.

## 1. Introduction

The ether-à-go-go-related gene (hERG) encrypts the voltage-gated potassium ion channel that plays a pivotal role in repolarizing the potential of cardiac action [1]. An impediment in the hERG channel may substantiate the risk of cardiac toxicity by retarding ventricular repolarization, which can be visualized explicitly through the extension of the time from ventricular depolarization to repolarization (QT interval) on electrocardiography [2]. In this regard, it is remarkable to note that antiarrhythmics represent the drugs with the highest potential risk of prolonging the QT interval. Furthermore, antihistamines and serotonin receptor agonists also bring about the prolongation of the QT interval, leading to withdrawal due to potential cardiotoxicity [3,4,5]. Hence, the hERG potassium channel has emerged as an annexed target against which the off-target inhibitory activities should be measured in the early stage of drug discovery to avoid side effects [6,7].

In accordance with the necessity for drug discovery, a variety of in vitro experimental methods for measuring the hERG-related cardiotoxicity have become available, including radioligand binding assays [8], patch clamp assays [9], and rubidium flux assays [10]. These experimental techniques have often been too ineffective to cope with a huge number of small molecules in the early stage of drug discovery [11]. Therefore, a reliable computational method for estimating the binding affinity of a drug candidate to hERG would be useful for prioritizing molecules in drug discovery. The development of such computational methods has been facilitated with the accumulation of chemical information about hERG channel blockers in public datasets. Several in silico tools to predict hERG liability have accordingly been developed using ligand-based methods [12,13,14,15,16] and structure-based simulation studies [17,18]. In particular, quantitative structure–activity relationship (QSAR) approaches have been most actively pursued because it became plausible to determine the functional relationship between hERG liabilities and numerical molecular descriptors [19,20,21,22]. Although most QSAR modeling studies adopted one- and two-dimensional (2D) features as individual molecular descriptors, it was demonstrated that the accuracy could be improved significantly by incorporating the 3D features of molecules in the dataset [23].

Since the establishment of the comparative molecular field analysis (CoMFA) model [24], 3D-QSAR methods have been applied in predicting the molecular binding affinities to the hERG potassium channel [25,26]. Although most numerical molecular descriptors used in 3D-QSAR models were too imperfect to predict various physicochemical properties with accuracy, replacing the descriptors prepared with empirical potential functions with quantum mechanical descriptors proved to enhance the predictive capability [27,28,29]. In this study, our goal was to develop a potent 3D-QSAR prediction model for hERG inhibitory activities using an artificial neural network (ANN) algorithm. By virtue of integrating a rigorous 3D geometrical alignment protocol with the quantum mechanical molecular descriptors, the experimental hERG inhibition data for a variety of molecules compared reasonably well with those calculated with the newly obtained 3D-QSAR prediction models. These computational methods are expected to be useful for virtual screening of hERG blockers in the early stage of drug discovery.

## 2. Results and Discussion

The entire molecular dataset involved a broad spectrum of organic compounds with varying sizes and inhibitory activities, such that the MWs and pIC_50_ values ranged from 250 to 600 amu and from 2.40 to 9.41, respectively. Therefore, a total of 490 organic compounds were divided into the seven subgroups according to MW to ensure that the 3D structural alignment process would be specific and relevant to each subgroup. Table 1 provides the breakdowns of the seven molecular subsets for which the 3D-QSAR models for hERG inhibitory activity were derived and validated separately. For a balanced representation for model training and validation, the number of molecular elements was kept consistent among the subsets, and then further subdivided into a 4:1 ratio for training and test sets.

Achieving an accurate 3D-QSAR prediction model relies critically on the precise 3D structural alignment of molecules within a dataset. Because only a small deviation from the perfect molecular superposition may cause a large error in predicting the physicochemical properties [30], 3D molecular alignment has been considered the most problematic bottleneck in 3D-QSAR modeling. While the majority of molecular alignment techniques involve superimposing similar chemical groups, there have been innovative approaches proposed to align entire molecular structures by leveraging the 3D distribution of physicochemical properties [31,32,33,34]. We used the alignment method termed AlphaQ [35], in which pairwise 3D structural alignments were carried out by optimizing the *E_ij_* values in relation to the template molecule. This method has distinct advantages over the conventional ones in handling structurally diverse molecules without identical chemical moiety because the calculation of *E_ij_* values on the fully quantum mechanical basis adds a layer of accuracy and sophistication to the approach. Figure 1 illustrates the outcomes of the 3D structural alignments within each molecular subset. The concentration of core structures in the central region across all seven cases suggests a consistent pattern in the alignment of molecules within each subset. The variations in sidechain orientations may provide valuable information about the structural diversity of the compounds. These 3D structural alignments cannot be scored quantitatively like the conventional atom-by-atom matching protocol because no common molecular core is present. Therefore, it would be desirable to assess the accuracy of the alignments with the predictive capabilities of 3D-QSAR prediction models derived from the optimized molecular atomic coordinates.

The reliability of the 3D-QSAR models in predicting molecular pIC_50_ values was validated based on their correlation with the corresponding experimental data. Briefly, the squared Pearson correlation coefficient for both the training set (*R*^2^*_train_*) and the test set (*R*^2^*_test_*) were used as metrics to assess the accuracy of the pIC_50_ prediction models. The mathematical expressions for these two statistical parameters are as follows:(1)$$R_{train}^{2}=1-\frac{\sum_{i=1}^{train}{\left(y_{i}-{\hat{y}}_{i}\right)}^{2}}{\sum_{i=1}^{train}{\left(y_{i}-{\bar{y}}_{train}\right)}^{2}}\;\mathrm{and}\;R_{test}^{2}=1-\frac{\sum_{i=1}^{test}{\left(y_{i}-{\hat{y}}_{i}\right)}^{2}}{\sum_{i=1}^{test}{\left(y_{i}-{\bar{y}}_{test}\right)}^{2}}$$

Here, $\bar{y}$ is the average of experimental pIC_50_ data while $y_{i}$ and ${\hat{y}}_{i}$ represent the experimental and calculated pIC_50_ data of molecule *i*, respectively. The summations in *R*^2^*_train_* and *R*^2^*_test_* parameters extend across the molecules in both the training and test sets, respectively.

In Figure 2, the linear correlation diagrams depict the relationship between the experimental pIC_50_ values and those computed using the 3D-QSAR models involving the *E_ij_*-based molecular alignments and the quantum mechanical ESP descriptors. The 3D-QSAR models for pIC_50_ prediction appear to converge successfully in all seven molecular subsets as can be inferred from the *R*^2^*_train_* value of 0.98 as the smallest. This indicates a successful optimization of weighting parameters using the ANN algorithm. It is noteworthy that this optimization holds true irrespective of the MW range in the training set. The contrast between the high similarity in *R*^2^*_train_* values and the extensive range in *R*^2^*_test_* parameters is quite intriguing. It suggests that while the model performs well on the training sets across different MW ranges, its predictive power varies when applied to the test sets. The worst prediction results are observed in Subset 3 (Figure 2c) and Subset 6 (Figure 2f), which include the molecules with MWs ranging from 351 to 400 and from 501 to 550 amu, respectively. Such relatively low *R*^2^*_test_* values in the two subsets may be understood on the grounds that Subset 3 and 6 contain the widest range of experimental pIC_50_, values including those lower than 3.5 as well as those higher than 9.1 (Table 1). Despite the potential imperfection in the molecular pIC_50_ datasets, the difference between the *R*^2^*_train_* and *R*^2^*_test_* parameters falls into 0.198 in all seven test cases. This implies that the issue of overtraining is substantially mitigated in the present 3D-QSAR prediction models for hERG inhibitory activity.

With respect to the predictive capability of the present 3D-QSAR prediction models for hERG inhibitory activities, it is worth noting that the *R*^2^*_test_* parameters for all seven molecular subsets are higher than those for predicting hERG inhibition using biomimetic HPLC measurements [36] and those of QSAR prediction models derived by operating machine learning algorithms on 2D pharmacophore descriptors [16]. The outperformance of the present 3D-QSAR prediction model is attributed most probably to the appropriateness of 3D structural alignments using the quantum mechanical *E_ij_* values, as the preparation of the optimal molecular atomic coordinates plays a crucial role in achieving an accurate 3D-QSAR model. The hERG pIC_50_ prediction models derived in this work also appear to outperform the conventional 3D-QSAR methods that involved 3D pharmacophore descriptors in terms of the *R*^2^*_test_* values [37]. This suggests that quantum mechanical ESP descriptors outperform the ensemble of 3D pharmacophore models for hERG binders

The performances of the 3D-QSAR models were further addressed with the external predictivity parameter (*r*^2^*_pred_*) that has been widely used for validating statistical prediction methods [38,39]. Mathematically, the *r*^2^*_pred_* parameter can be expressed as follows:(2)$$r_{pred}^{2}=1-\frac{\sum_{i=1}^{test}{\left(y_{i}-{\hat{y}}_{i}\right)}^{2}}{\sum_{i=1}^{test}{\left(y_{i}-{\bar{y}}_{train}\right)}^{2}}$$

Here, $y_{i}$ and ${\hat{y}}_{i}$ denote the experimental and calculated data for the molecules in the test set, while ${\bar{y}}_{train}$ is the averaged value for molecules in the training set. The *r*^2^*_pred_* parameter has an advantage over the corresponding *R*^2^*_test_* parameter in the context that characteristics of the training set are also reflected in evaluating a prediction model as well as those of the test set. As shown in Figure 2, the *r*^2^*_pred_* parameters associated with predicting hERG pIC_50_ values range from 0.758 to 0.880 among the seven training and test sets, which exceed the threshold (0.6) for the qualification of a statistical prediction model [38]. This confirms the reliability of the present 3D-QSAR models for predicting hERG inhibitory activities. It is also noteworthy that the disparity between the *r*^2^*_pred_* and *R*^2^*_test_* values is negligible (less than 5%) in Subsets 3–7, implying that the training and test sets were divided reasonably well in coping with the molecules with MWs larger than 350 amu. In contrast, the predictive capability was affected significantly by the compositions of training and test sets in Subsets 1 and 2, as can be inferred from the relatively large differences between the *r*^2^*_pred_* and *R*^2^*_test_* values (Figure 2). Overall, both statistical validation parameters support the reliability of the present 3D-QSAR models in predicting the molecular pIC_50_ values of hERG blockers.

The reasonably good predictive capability of the present 3D-QSAR model may also be elucidated in the context that 3D distribution of quantum-mechanically calculated ESP would be superior to classical 1D and 2D molecular properties as numerical descriptors [35]. The suitability of quantum mechanical ESP distribution as a numerical molecular descriptor was also demonstrated in estimating the potencies of ice recrystallization inhibitors [40]. The 3D ESP descriptors developed in this study differ from those in other research, as the ESP values were computed at every 3D grid point within a shared box encompassing all molecules in the dataset, rather than focusing solely on surface points. This modification is actually necessary to obtain an accurate 3D-QSAR prediction model for complicated biological properties such as hERG inhibitory activity. Such fully quantum mechanical ESP distribution may also serve as an effective numerical molecular descriptor to derive other 3D-QSAR prediction models for a variety of biochemical and pharmacological properties.

Although the calculated pIC_50_ values of some molecules deviated substantially from the experimental counterparts (Figure 2), it was difficult to further enhance the predictive capability of 3D-QSAR models either by upgrading the quantum chemical methods for preparing the ESP descriptors or by increasing the number of hidden layers in ANN parameterizations. The largest errors in hERG pIC_50_ prediction are observed for CID11692293 and CID71720519 (Figure 3), with the absolute errors of 1.56 and 1.52, respectively. The subpar prediction outcomes illustrate that altering just a few molecules in the dataset can significantly impact the performance of a QSAR model [41]. If CID11692293 is excluded in the dataset, for instance, the *R*^2^*_test_* parameter of Subset 6 increases significantly from 0.808 to 0.915. With respect to the poor predictive capabilities of the two molecules, it is noteworthy that both CID11692293 and CID71720519 contain a tertiary amine moiety that must be partially protonated under physiological conditions. Therefore, the large errors in the calculated pIC_50_ values of the two molecules may stem from neglecting the contributions of the protonated form to 3D structural alignments as well as to ESP descriptors. It can thus be argued that the accuracy of a 3D-QSAR prediction model would increase through proper modeling of molecular hydrophobicity and hydrophilicity. In this regard, the use of molecular conformations derived through consideration of solvation effects would be more desirable than those obtained with quantum chemical calculations in vacuo.

Similar to other 3D-QSAR methods, it is a limitation of the present hERG pIC_50_ prediction models that only a single conformation of a molecule can be used both in the calculation of ESP descriptors and in 3D structural alignments. This restraint seems to cause an error in predicting the hERG pIC_50_ values due to the imperfection of the 3D-QSAR models. We note in this regard that CID11692293 and CID71720519 have seven and six rotatable bonds, respectively, indicating the presence of multiple conformational degrees of freedom. Nonetheless, only one conformer was taken into account in predicting hERG inhibitory activities on the grounds that its potential energy calculated at the RHF/6-31G** level corresponded to a local energy minimum. A large error can therefore be accumulated in the predicted hERG pIC_50_ values because the contributions of other conformational isomers were excluded during the derivation and the validation of the final 3D-QSAR models. In a strict sense, the enumeration of all molecular conformations is necessary to derive accurate 3D-QSAR prediction models because the experimental data utilized in constructing the model were measured, taking into account all torsional degrees of freedom. To enhance the predictive capability of a 3D-QSAR model for hERG inhibitory activities, therefore, it is required to reflect the contributions of multiple conformers of each dataset molecule both in ESP descriptor calculations and in 3D structural alignments.

The error accumulation problem may become severe when the dataset involves a number of molecules possessing high conformational degrees of freedom. In this case, implementing the 4D-QSAR formalism to calculate molecular descriptors, considering the conformational diversity of individual molecules, would enhance the predictive capability [42]. Because a variety of simulation methods for rigorous conformational sampling are available in the literature, our future research will aim to enhance the performance of the hERG pIC_50_ prediction model within the 4D-QSAR framework using an advanced graphics processing unit architecture.

## 3. Materials and Methods

### 3.1. Preparation of the Molecular Dataset for hERG Channel Binders

Although the accuracy of 3D-QSAR models depends critically on the structural alignments among the molecules, it is very difficult to align the 3D molecular structures in appropriate directions, especially when the dataset involves a broad range of molecular weight (MW) [43]. The difficulty in achieving an accurate structural alignment is ascribed in a large part to the ambiguity in selecting a prototypical molecule that has to serve as the template to align all the other molecules. The alignment errors would be ameliorated if a dataset contained the molecules with similar MWs [41]. Therefore, the entire molecular dataset was divided into seven subsets with MW ranges of 251–300, 301–350, 351–400, 401–450, 451–500, 501–550, and 551–600 atomic mass unit (amu). Individual subsets were then filled with 70 molecular datapoints for the half-maximal inhibitory concentration (IC_50_) against the hERG potassium channel, which were extracted at random from the dataset used in developing the artificial intelligence method for topology-inferred drug addiction learning [44]. A total of 490 experimental IC_50_ datapoints for the molecules with a variety of atomic compositions, shapes, and sizes were thus used to derive and validate the seven 3D-QSAR prediction models, which were adequate for a certain MW range. PubChem CID’s, molecular weights, and experimental and calculated pIC_50_ values of all the molecules in the dataset were provided in Appendix A. For simplicity, the experimental IC_50_ values expressed in molar concentration were converted to the numbers given by taking the negative decadic logarithm (pIC_50_). All seven molecular subsets were then subdivided into training and test set at the ratio of 56:14 to construct a 3D-QSAR model and to validate the predictive capability, respectively.

### 3.2. Pairwise 3D Structural Alignments of the Molecules in the Dataset

To prepare the starting point for structural alignments, 3D structures of all the molecules in the dataset were fully optimized via quantum chemical calculations at the RHF/6-31G** level of theory. These preliminary calculations were carried out using Gaussian09 program on Linux desktop 64-bit platforms. The molecule with the highest MW in a subset was selected as the template for the multiple pairwise structural alignments in the common 3D grid box. Three-dimensional atomic coordinates of all the other molecules in a subset were determined with respect to the template molecule whose position was fixed in the grid box. The dimension of the grid box that was common to the molecules in a subset were set equal to the maximal distances along the three coordinate axes of the van der Waals volumes of individual molecules. During the 3D structural alignments, the marginal distance of 2.7 Å was appended to the length, width, and height of the common grid box to ensure enough space for translational and rotational movements. This 3D grid box was completed via the uniform spacing of grid points at 0.106 Å along the three axes.

Translating and rotating each molecule (target) to maximize the overlap with the template molecule was a key step in the pairwise structural alignments. For each target molecule, a total of 2000 rotamers along the three axes were taken into account to determine the optimal atomic coordinates with respect to the template. The Hopf fibration method [45] for sampling in the SO(3) rotation group was used as a systematic way to cover the rotational degrees of freedom. It was used as a strategy for saving computational cost, as the charge density distribution of a molecule was calculated only once for the starting structure, whereas those of the rotamers were interpolated at each grid point.

Using 2000 rotamers for a target molecule (*j*) along with the charge density distribution in the 3D grid box, the optimal structural alignment with the template molecule (*i*) was searched exhaustively by systematically translating each rotamer. These translational shifts were iterated by changing the displacement vectors until the quantum mechanical cross correlation (*E_ij_*) between *i* and *j* reached the maximum. *E_ij_* was defined using the electrostatic potential (ESP) of *i* (*ϕ_i_*(x, y, z)) and the charge density of *j* (*ρ_j_*(x, y, z)).
(3)$$E_{ij}={∭}_{V}\varphi_{i}(x,y,z)\rho_{j}(x,y,z)dV$$

In terms of molecular interactions, *E_ij_* typically represents the energy associated with the repulsive electrostatic interactions between molecule *i* and *j*. All *E_ij_* values were calculated via the fast Fourier transform algorithm [46]. The optimal alignment for *j* was determined by selecting the rotamer with the highest *Eij* value, and this selected configuration was then employed as input for computing the molecular ESP descriptor.

### 3.3. Calculations of the 3D Molecular Descriptors

The three-dimensional distribution of ESP surrounding a molecule, which harbors 2n electrons, was derived from its determinantal wavefunction. This comprised n molecular orbitals calculated using an ab initio quantum chemical method at the RHF/6-31G** level. By employing the individual molecular wavefunctions, charge density (*ρ*) values were computed at every 3D grid point positioned with uniform spacing of 0.212 Å within a shared rectangular box. The ESP (*ϕ*) value at each grid point was ascertained through the solution of Poisson’s equation.
(4)$${\vec{\nabla }}^{2}\varphi (x,y,z)=\rho (x,y,z)\ldots$$

It might be a technically sound approach to prepare a numerical molecular descriptor in the form of a K-dimensional vector comprising the ESP values at the K grid points in the common 3D grid box. Owing to a large number of grid points (K = 1,191,016), it was reasonable to reduce the dimensionality so as to be adequate for QSAR modeling. The principal component analysis (PCA) would be effective in this case, which has been widely used to extract essential information from high-dimensional numerical data while simplifying representation [47,48]. We used these reduced molecular ESP descriptors to derive 3D-QSAR prediction models for the activities of hERG blockers through the ANN algorithm. It was intriguing that these descriptors, derived from fully quantum mechanical calculations, were expected to outperform conventional descriptors in terms of correlation with experimental data.

### 3.4. Derivation of the Prediction Models for the Activities of hERG Blockers

Deriving a 3D-QSAR model for predicting pIC_50_ values of hERG channel blockers using advanced computational protocols was a commendable effort. This was made possible by the adoption of the ANN algorithm with a feed-forward architecture and backpropagation of error network [49]. The network included input, hidden, and output layers, each serving a specific role in the prediction process, as depicted in Figure 4. The input layer had 56 neurons representing the projected ESP vectors of the training-set molecules. All these input neurons (${\hat{I}}_{k}$s) were then processed using a sigmoidal function after multiplying the weighting factors (*w_ki_*’s) to form the hidden layer with 35 intermediate neurons (${\hat{H}}_{i}$s). Similarly, ${\hat{H}}_{i}$s were combined in turn to define a single output neuron ($\hat{O}$) that consisted of the predicted pIC_50_ values of *N* molecules in the training set.
(5)$${\hat{H}}_{i}=sgm\left(\sum_{k=1}^{N}w_{ki}{\hat{I}}_{k}\right)\;\mathrm{and}\;\hat{O}=sgm\left(\sum_{i=1}^{M}w_{ij}{\hat{H}}_{i}\right)$$

Here, sgm(x) denotes the sigmoidal function given by (1 + e^−x^)^−1^. The output neuron can therefore be expressed with the input vectors as follows:(6)$$\hat{O}=sgm\left(\sum_{i=1}^{M}w_{ij}sgm\left(\sum_{k=1}^{N}w_{ki}{\hat{I}}_{k}\right)\right)$$

The optimization of the 3D-QSAR model for pIC_50_ prediction could be simplified by limiting the number of neurons in the hidden layer (M) to 35. To facilitate the whole training process, the experimental pIC_50_ values were normalized to a range of 0 to 1 to be processed with the sigmoidal function. The 3D-QSAR prediction models were thus trained on a consistent and standardized scale by using the normalized experimental pIC_50_ values. Finally, the model building proceeded via a gradient-based minimization on the error hypersurface (*F*), given by the sum of the square differences between the experimental (*D_j_*) and the estimated (*O_j_*) pIC_50_ values of *N* molecules in the training set.
(7)$$F=\sum_{j=1}^{N}{\left(D_{j}-O_{j}\right)}^{2}$$

The F value of 10^−4^ was used as the criterion for the convergence of weighting parameters.

## 4. Conclusions

To obtain a reliable computational tool for estimating molecular hERG inhibitory activities, the QSAR prediction models were derived using the 3D distribution of quantum mechanical ESP values as the mathematical molecular descriptors. It is a strategic move to enhance the predictive capability of the QSAR models by carrying out the pairwise 3D molecular structural alignments by maximizing the quantum mechanical cross correlations between the template and other molecules in the dataset. This alignment protocol demonstrated merit compared to the conventional atom-by-atom matching method. It was effective in handling structurally diverse molecules with the same rigor as chemical derivatives sharing an identical scaffold. Nonetheless, the ambiguity in determining the optimal structural alignments between small and large molecules made it difficult to derive an accurate 3D-QSAR prediction model. This problematic alignment bottleneck was alleviated to a substantial degree by dividing the dataset molecules into seven subsets, each of which contained the molecules with similar MWs. Consequently, highly predictive QSAR models were obtained for all seven molecular subsets, indicating that the pairwise 3D structural alignments and the quantum mechanical ESP descriptors would be appropriate to develop QSAR prediction models for hERG inhibitory activities. Given their high predictive capability and the simplicity of model development, the 3D-QSAR prediction models developed in this study are anticipated to function as an effective virtual screening tool for potential cardiotoxicity.

## Figures and Tables

**Figure 1 pharmaceuticals-16-01509-f001:**
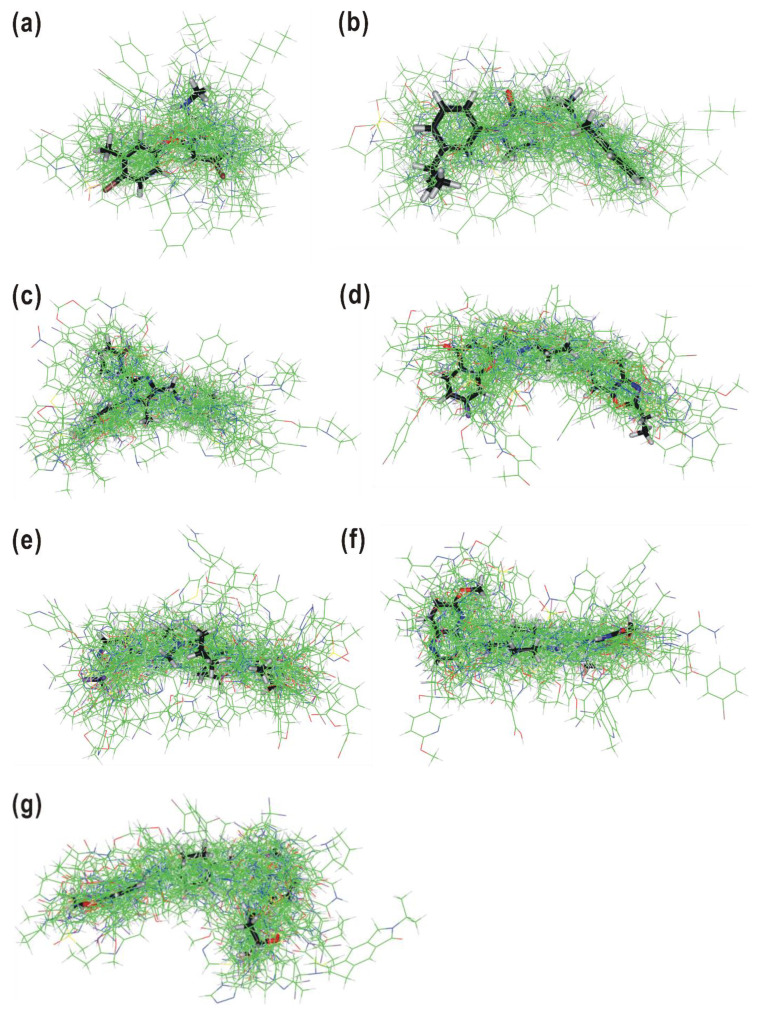
Outcomes of 3D structural alignments within the molecules of (**a**) Subset 1, (**b**) Subset 2, (**c**) Subset 3, (**d**) Subset 4, (**e**) Subset 5, (**f**) Subset 6, and (**g**) Subset 7. Carbon atoms of the template and target molecules are denoted in black and green, respectively.

**Figure 2 pharmaceuticals-16-01509-f002:**
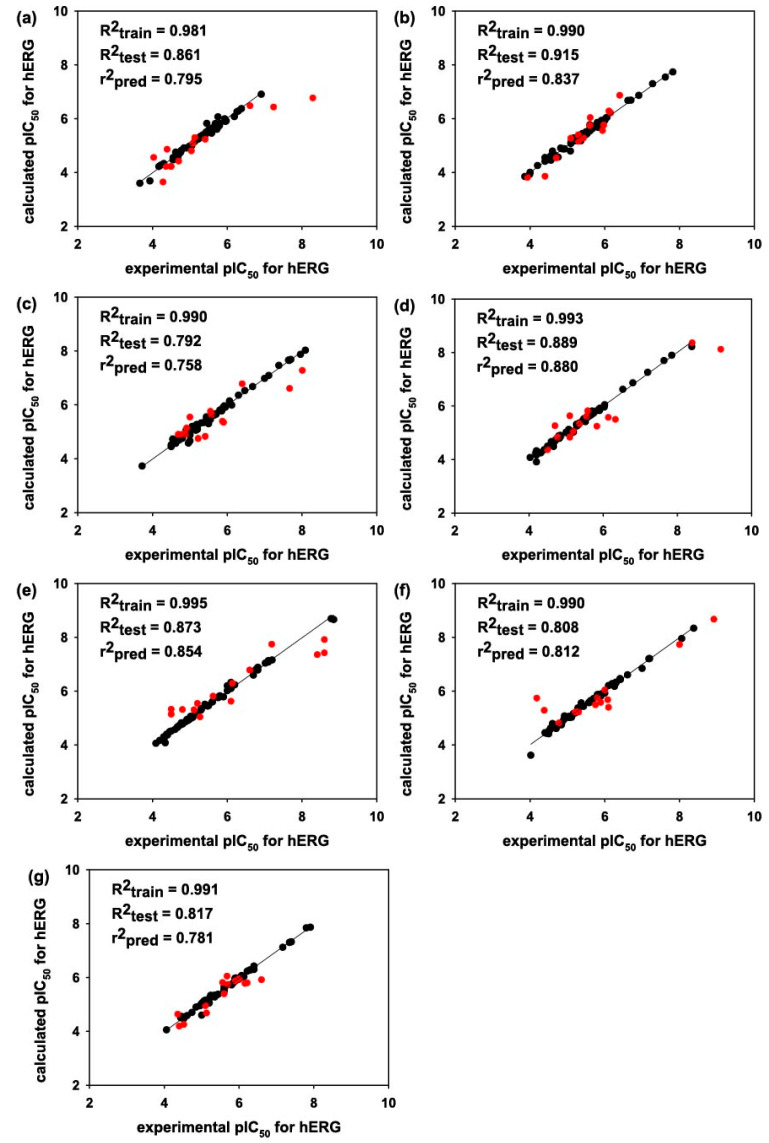
Correlation diagrams illustrating the relationship between experimental and calculated hERG pIC_50_ values for (**a**) Subset 1, (**b**) Subset 2, (**c**) Subset 3, (**d**) Subset 4, (**e**) Subset 5, (**f**) Subset 6, and (**g**) Subset 7. Molecules in the training set are marked with black circles, while those in the test set are highlighted with red circles.

**Figure 3 pharmaceuticals-16-01509-f003:**
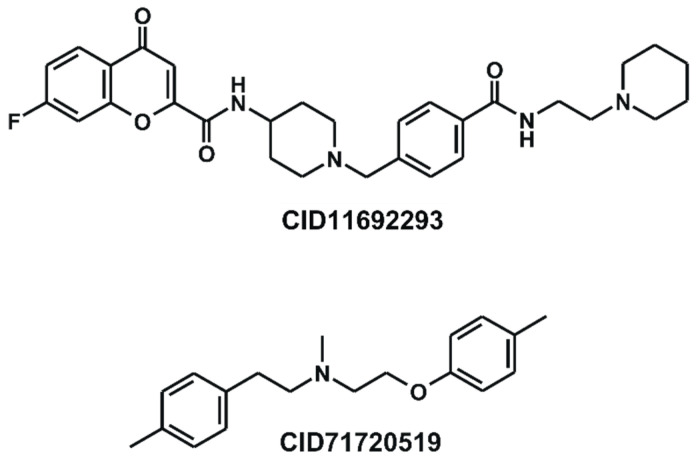
Chemical structures of CID11692293 and CID71720519.

**Figure 4 pharmaceuticals-16-01509-f004:**
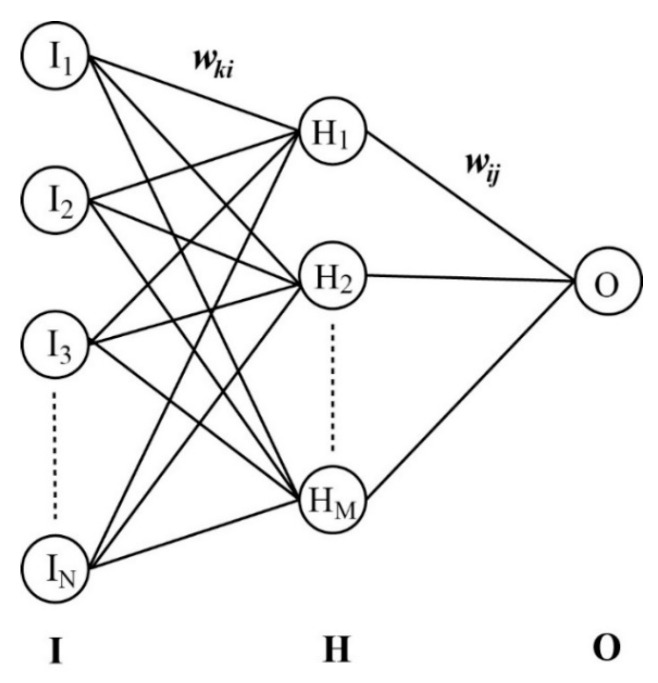
Schematic diagram of N × M × 1 neural network to derive a 3D-QSAR model for predicting the pIC_50_ data of hERG blockers. Column I, H, and O denote the input, hidden, and output layer, respectively. Neurons in these three layers are interconnected through the weighting matrices *w_ki_* and *w_ij_*.

**Table 1 pharmaceuticals-16-01509-t001:** Attributes of the seven molecular subsets employed in establishing and validating a 3D-QSAR prediction model for hERG inhibitory activity.

Molecular Subset	MW Range	pIC_50_ Range	No. of Training-Set Molecules	No. of Test-Set Molecules
Subset 1	250–300	3.66–8.29	56	14
Subset 2	301–350	3.86–8.82	56	14
Subset 3	351–400	3.49–9.12	56	14
Subset 4	401–450	4.03–9.17	56	14
Subset 5	451–500	4.05–9.06	56	14
Subset 6	501–550	2.40–9.41	56	14
Subset 7	551–600	4.06–8.77	56	14

## Data Availability

The data can be shared on request.

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
