# Peer review of "Derivation of Highly Predictive 3D-QSAR Models for hERG Channel Blockers Based on the Quantum Artificial Neural Network Algorithm"

_pharmaceuticals, 2023, doi:10.3390/ph16111509_

Round 1

Reviewer 1 Report

The authors presented a paper in which the distribution of quantum–mechanical electrostatic potential (ESP) is treated as a molecular descriptor. Then the QSAR analysis was performed to estimate the inhibitory activity against the hERG potassium channel. In the beginning, the authors clearly highlighted the importance of the ether-á-go-go-related gene (hERG) in drug discovery. The purpose of the work and the possibilities of potential use of the results were also clearly marked. Thanks to the 3D-QSAR model, it was possible to predict the pIC50 values of hERG channel blockers.

Page 4, line 168 – what does Figure 35 mean? What is it about or referring to? What does the phrase “during the optimization……” mean?

The predicted pIC50 values for hERG correlate very well with the experimentally determined values. It confirms that 3D-QSAR could be useful for determining IC50 and also for other classes of compounds using the value of quantum–mechanical electrostatic potential. The authors also described the limitations of the method. I believe that the authors should consider removing Figure 4. These relationships are already presented in the previous Figures (3a-3g) and are better visible there. In my opinion, there is no need to repeat the charts, especially since the figure is not very legible.

After minor changes, I recommend the work for printing.

Author Response

1) Page 4, line 168 – what does Figure 35 mean? What is it about or referring to? What does the phrase “during the optimization……” mean?

The first sentence in line 168 was wrongly written in the original manuscript. It has been corrected to “The optimization of the 3D-QSAR model for pIC50 prediction could be simplified by limiting the number of neurons in the hidden layer (M) to 35.” (Line 166 in the revised manuscript).

2) The predicted pIC50 values for hERG correlate very well with the experimentally determined values. It confirms that 3D-QSAR could be useful for determining IC50 and also for other classes of compounds using the value of quantum–mechanical electrostatic potential. The authors also described the limitations of the method. I believe that the authors should consider removing Figure 4. These relationships are already presented in the previous Figures (3a-3g) and are better visible there. In my opinion, there is no need to repeat the charts, especially since the figure is not very legible.

Following the suggestion, we have removed Figure 4 and the third paragraph on p. 9 in the revised manuscript.

Author Response

1) The calculation is done at a relatively low level, RHF/6-31G**. Since conformational optimization of 490 molecules has been done, it is understandable that a low level would be used. Probably using these optimized structures is adequate, given the nature of the computation. However, it would be useful to know whether, at least for one or two subsets, using a higher level (B3LYP/6-311+G**, perhaps) single point computation with the optimized structures would give a better potential for the calculation of Eij. Single point computations are not terribly expensive, even with a high-level method, as the molecules are not large. In tracking down the remaining error, this should either be something that has to be taken into account, or else eliminated as something that does not need to be taken into account. Even the largest molecules have MW < 550; this should not be too difficult, especially for subsets 1 and 2 with even smaller molecules.

We agreed that the performance of 3D-QSAR prediction model would be enhanced when the structural alignments were carried out with a higher level of theory than RHF/6-31G**. However, it will take at least a couple of months to recalculate the 3D structural alignments even in the case of Subset 1 because the common grid box contains a huge number of grid points (K=1,191,016). Therefore, we would like to limit the scope the present paper to the effectiveness of a fully quantum chemical method for deriving an accurate 3D-QSAR prediction model without considering the dependence of the predictive capability on the theoretical level of calculations.

2) The authors correctly note that using a single conformation may produce a poor result; using the single optimized structure is equivalent to doing the calculation at 0 K. There are other problems, of which probably the most serious is the absence of water, almost always involved in a binding pocket of a channel. However, if a single conformation with a higher-level potential gives an appreciably better result, perhaps this problem would turn out to be unimportant. The Fig. 5 molecules have a problem with possible ionization. It would be useful to know whether there is a difference in the accuracy with molecules that are hydrophobic and hydrophilic. If this takes a major effort, the absence of testing for hydrophobicity can just be noted as something that will be done in future work. If it can be done with not too much effort, it would add to the value of this paper.

We also agreed that the accuracy of a 3D-QSAR prediction model would increase further by proper modeling of molecular hydrophobicity and hydrophilicity. In this regard, the use of molecular conformations derived under consideration of solvation effects would be more desirable than those obtained with quantum chemical calculations in vacuo. To explain these, we have added two sentences on p. 9 lines 314-318 in the revised manuscript.

3) The method is given, but the software used is not stated: Gaussian? NWChem? Whatever it is, it should be stated, and a reference given. Also, there is no acknowledgement of a computer facility. Was all this done on a lab desktop, or was a major computer facility used? If this could be done on a desktop, it must have taken quite a while; if an external facility was used, it must be acknowledged.

In accordance with the comment, we have clarified that all quantum chemical calculations were carried out using Gaussian09 program on Linux desktop 64-bit platforms (p. 3 line 97-98 in the revised manuscript). The reference was omitted because Gaussian is a commercial software package.

4) Out of 50 references, only 7 are 2020 or later. I have not done a literature search to find later references, but perhaps the authors should.

Following the suggestion, we have updated the five references (Refs. 1, 4, 9, 37, and 49) in the revised manuscript.

Minor matters

a) On line 270, Fig 3 is supposed to show something about r2pred; I don’t see this.

The r2pred parameters have been presented explicitly in Figure 3 along with R2train and R2test values.

b) On line 318, “neither” should be “either” On the whole, this work is a possibly useful step forward, and with further development should be of practical importance.

The word has been corrected as indicated by Reviewer.